# Representing Long-Range Context for Graph Neural Networks with Global Attention

**Zhanghao Wu**[*1], **Paras Jain**[*1], **Matthew A. Wright**[1]
**Azalia Mirhoseini**[2], **Joseph E. Gonzalez**[1], **Ion Stoica**[1]
[1]UC Berkeley,    [2]Google Brain
Correspondence to: {zhwu, paras_jain}@berkeley.edu
[*]equal contribution, determined via a random coin flip.

## Abstract

Graph neural networks are powerful architectures for structured datasets. However, current methods struggle to represent long-range dependencies. Scaling the depth or width of GNNs is insufficient to broaden receptive fields as larger GNNs encounter optimization instabilities such as vanishing gradients and representation oversmoothing, while pooling-based approaches have yet to become as universally useful as in computer vision. In this work, we propose the use of Transformer-based self-attention to learn long-range pairwise relationships, with a novel "readout" mechanism to obtain a global graph embedding. Inspired by recent computer vision results that find position-invariant attention performant in learning long-range relationships, our method, which we call `GraphTrans`, applies a permutation-invariant Transformer module after a standard GNN module. This simple architecture leads to state-of-the-art results on several graph classification tasks, outperforming methods that explicitly encode graph structure. Our results suggest that purely-learning-based approaches without graph structure may be suitable for learning high-level, long-range relationships on graphs. Code for `GraphTrans` is available at https://github.com/ucbrise/graphtrans.

## 1   Introduction

Graph neural networks (GNNs) enable deep networks to process structured inputs such as molecules or social networks. GNNs learn mappings that compute representations at graph nodes and/or edges from the structure of and features in their neighborhoods. This neighborhood-local aggregation leverages the relational inductive bias encoded by the graph's connectivity [3]. Similar to convolutional neural networks (CNNs), GNNs can aggregate information from beyond local neighborhoods by stacking layers, effectively broadening the GNN receptive field.

However, GNN performance drops dramatically when its depth increases [21]. This limitation has hurt the performance of GNNs on whole-graph classification and regression tasks, where we want to predict a target value describing the whole graph that may rely on long-range dependencies that may not be captured by a GNN with a limited receptive field [35]. Consider for example a large graph where node $A$ must attend to a distant node $B$ which is $K$-hops away. If our GNN layer aggregates only over a node's one-hop neighborhood, then a $K$-layer GNN is required. However, the width of the receptive field of this GNN will grow exponentially, diluting the signal from node $B$. That is, simply expanding the receptive field to a $K$-hop neighborhood may not capture these long-range dependencies either [40]. Often, "too deep" GNNs lead to node representations that collapse to be equivalent over the entire graph, a phenomenon sometimes called *oversmoothing* or *oversquashing* [21, 5, 2]. Therefore, the maximum context size for common GNN architectures is effectively limited.

35th Conference on Neural Information Processing Systems (NeurIPS 2021).

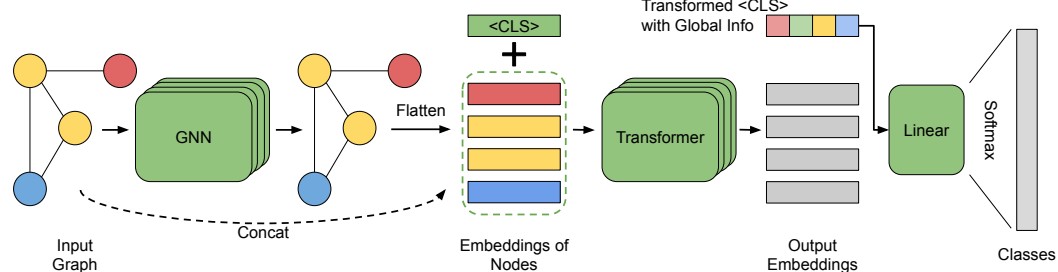

Figure 1: Architecture of `GraphTrans`. A standard GNN submodule learns local, short-range structure, then a global Transformer submodule learns global, long-range relationships.

Several proposed methods combat the oversmoothing problem via intermediate pooling operations similar to those found in today's CNNs. Graph pooling operations gradually coarsen the graph in progressive GNN layers, usually by collapsing neighborhoods into single nodes [9, 37, 20, etc.]. In theory, hierarchical coarsening should allow better long-range learning, both by reducing the distance information has to travel and by filtering out unimportant nodes. However, no graph pooling operation has been found that is as universally applicable as CNN pooling. State-of-the-art results are often obtained with models using no intermediate graph coarsening [27], and some results suggest neighborhood-local coarsening may be unnecessary or counterproductive [23].

In this work, we take a different approach at graph pooling and learning long-range dependencies in GNNs. Like hierarchical pooling, our method is also inspired by methods for computer vision: we replace some of the atomic operations that explicitly encode relevant relational inductive biases (i.e., convolutions or spatial pooling in CNNs, neighborhood coarsening in GNNs) with purely learned operations like attention [11, 4, 7].

Our method, which we call Graph Transformer (`GraphTrans`, see Fig. 1), adds a Transformer subnetwork on top of a standard GNN layer stack. This Transformer subnetwork explicitly computes all pairwise node interactions in a position-agnostic fashion. This approach is intuitive as it retains the GNN as a specialized architecture to learn local representations of the structure of a node's immediate neighborhood while leveraging the Transformer as a powerful global reasoning module. This parallels recent computer vision architectures, where authors have found hard relational inductive biases important for learning short-range patterns but less useful or even counterproductive in modeling long-range dependencies [25]. As the Transformer without a positional encoding is permutation-invariant, we find it is a natural fit for graphs. Moreover, `GraphTrans` does not require any specialized modules or architectures and can be implemented in any framework atop any existing GNN backbone.

We evaluate `GraphTrans` on a variety of popular graph classification datasets. We find significant improvements in accuracy on OpenGraphBenchmark [15] where we achieve state-of-the-art results on two graph classification tasks. Moreover, we find substantial improvements on the molecular dataset NCI1. Surprisingly, we find our simple model outperforms complex baselines for long-range modeling in graphs via hierarchical clustering such as self-attention pooling [20].

Our contributions are as follows:

- We show that long-range reasoning via Transformers improve graph neural network (GNN) accuracy. Our results suggest that modelling all pairwise node-node interactions in the graph is particularly important for large graph classification tasks.

- We introduce a novel GNN "readout module." Inspired by text-classification applications of Transformers, we use a special "<CLS>" token whose output embedding aggregates all pairwise interactions into a single classification vector. We find that this approach outperforms both non-learned readout methods like global pooling as well as learned aggregation methods like graph-specific pooling methods [37, 20] and "virtual node" approaches.

- Using our novel architecture `GraphTrans`, we obtain state-of-the-art results on several OpenGraph-Benchmark [15] datasets and the NCI biomolecular datasets [30].

## 2   Related Work

**Graph Classification.**   Graph classification is an important task in real-world applications. Though GNNs encode the structured data into the node representations, aggregation of the representations to a single graph embedding for graph classification is still a problem. Similar to CNNs, pooling in GNNs can be either global, reducing a set of node and/or edge encodings to a single graph encoding, or local, collapsing subsets of nodes and/or edges to create a coarser graph. Paralleling the use of intermediate pooling within CNNs, several authors have proposed local pooling operations meant to be used within the GNN layer stack, progressively coarsening the graph. Methods proposed include both learned pooling schemes [37, 20, 14, 16, 1, etc.] and non-learned pooling methods based on classic graph coarsening schemes [10, 9, etc.]. However, the effectiveness or necessity of hierarchical, coarsening-based pooling in GNNs is unclear [23]. On the other hand, the most common global, whole-graph pooling methods, are i) non-learned mean or max-pooling over nodes and ii) the "virtual node" approach, where a final GNN layer outputs an embedding for a single virtual node that is connected to every "real" node in the graph.

A notable work related to graph pooling is the DAGNN (Directed Acyclic Graph Neural Network) of Thost and Chen [27], which had obtained the previous state-of-the-art accuracy on OGBG-Code2. The DAGNN layer aggregates over the entire graph within each layer via an RNN that traverses the DAG, unlike most GNN layers that only aggregate over a node's neighborhood. While they did not characterize this method as a pooling operation, it is similar to `GraphTrans` in that it acts as a learned global pooling (in that it aggregates the embeddings of every node in a DAG into the sink nodes) that can model long-range dependencies. Note that `GraphTrans` is also complementary to DAGNN because their final graph-level pooling operation is a global max-pooling over the sink nodes rather than a learned operation.

**Transformers on Graphs.**   Several authors have investigated applications of Transformer architectures to graphs. Recent works such as Zhang et al. [38], Rong et al. [24], and Dwivedi and Bresson [12] propose GNN layers that let nodes attend to other nodes in some surrounding neighborhood via Transformer-style attention, whereas we use self attention for a permutation-invariant, graph-level pooling or "readout" operation that collapses node encodings to a single graph encoding. Of these, Zhang et al. [38] and Rong et al. [24] tackle the problem of learning long-range dependencies without over smoothing by allowing nodes to attend to more than just the one-hop neighborhood: Zhang et al. [38] take the attended neighborhood radius as a tuning parameter and Rong et al. [24] attend to neighborhoods of random size during training and inference. In contrast, we use whole-graph self-attention to allow for learning of long-range dependencies.

While Zhang et al. [38] do not consider whole-graph prediction problems, in the case of Dwivedi and Bresson [12], when a graph-wide embedding was needed for graph classification or regression, they used global average pooling over the nodes, while Rong et al. [24] take a weighted sum over nodes with the weights computed bypassing the $h_v^L$'s to a two-layer MLP. Note also that prior works consider graph-specific versions of a Transformer's positional encoding, while we omit positional encodings to ensure permutation invariance.

**Efficient Transformers.**   Transformer [28] has been widely used in sequence modeling. Recently, modifications of the transformer architecture emerge to further improve the efficiency [34, 19, 6]. The LiteTransformer [34] with less FLOPs, Reformer [19] with complexity, and Performer [6] with both less computation and memory complexity. Neural architecture search (NAS) was also applied to Transformer to fulfill the resource constraints for the edge devices [32]. These off-the-shelf architectures are orthogonal to our `GraphTrans` and can be adopted to improve the scalability.

## 3   Motivation: Modeling Long-Range Pairwise Interactions

To summarize, attempting long-range learning on graphs via stacking GNN layers or hierarchical pooling have not yet led to performance increases, and while some works have shown some success in expanding the receptive field of a single GNN layer beyond a one-hop neighborhood [38, 24, 40], it remains to be seen how this approach will scale to very large graphs with thousands of nodes.

An inspiration for an alternative approach can be found in the recent computer vision literature. In the last few years, researchers have found that attention mechanisms can act as drop-in replacements

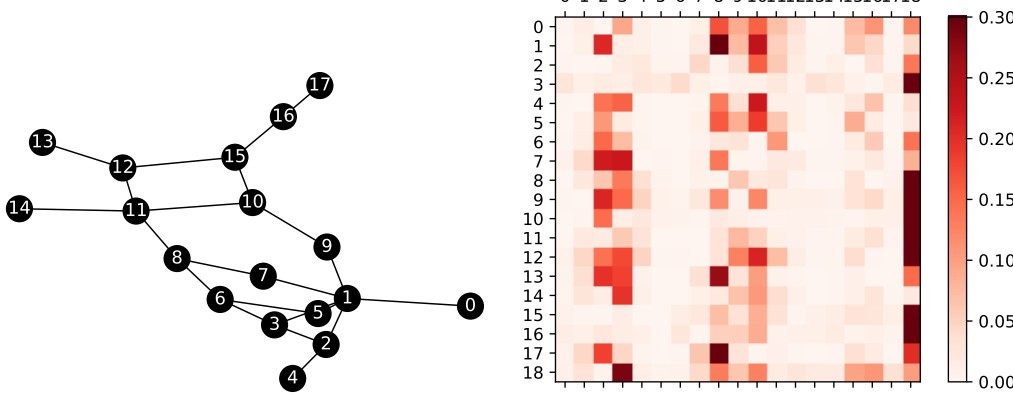

(a) Example graph from Code2.

(b) Corresponding attention map from `GraphTrans`.

Figure 2: Example graph and attention map in our `GraphTrans`. The graph is randomly sampled from the Code2 validation set. The attention map is retrieved from the first layer of the transformer module in our `GraphTrans`. The horizontal axis corresponds to targets and the horizontal axis corresponds to sources (so, attention weights will sum to one over the horizontal axis). Note that in (b), index 18 corresponds to the special <CLS> token described in section 4.

for traditional CNN convolutions [4, 7]: attention layers can learn to reproduce the strong relational inductive biases induced by local convolutions. More recently, state-of-the-art approaches to several computer vision tasks use an attention-style submodule on top of a traditional CNN backbone [2, 33, etc.]. These results suggest that while strong relational inductive biases are helpful for learning local, short-range correlations, for long-range correlations less structured modules may be preferred [2].

We leverage this insight to the graph learning domain with our `GraphTrans` model, which uses a traditional GNN subnetwork as a backbone, but leaves learning long-range dependencies to a Transformer subnetwork with no graph spatial priors. As mentioned, our Transformer application lets every node attend to every other node (unlike other approaches of applying Transformers to graphs that only allow attention to neighborhoods), which incentivizes the Transformer to learn the *most important* node-node relationships, instead of favoring nearby nodes (the latter task having been offloaded to the preceding GNN module).

Qualitatively, this scheme provides evidence that long-range relationships are indeed important. An example application of `GraphTrans` on the OGB Code2 dataset is depicted in Figure 2. In this task, we take in the Abstract Sentence Tree obtained by parsing a Python method and need to predict the tokens that form the method name. The attention map exhibits similar patterns to those found in NLP applications of Transformers: some nodes receive significant weighting from many other nodes, regardless of the distance between them. Note that node 17 assigns significant importance to node 8, despite these two nodes being five hops away. Also, in Figure 2's attention map, index 18 refers to the embedding corresponding to the special <CLS> token we use as a readout mechanism, described in more detail below. We allow this embedding to be learnable, so the many nodes attending to it (represented by the many dark cells in column 18) may suggest these nodes are obtaining some graph-general memory from the learned embedding. This qualitative visualization, along with our new state-of-the-art results, suggest that removing spatial priors when learning long-range dependencies may be necessary for effective graph summarization.

## 4    Learning Global Information with `GraphTrans`

Referring back to Figure 1, `GraphTrans` consists of two primary modules: a GNN subnetwork followed by a Transformer subnetwork. We discuss these in detail next.

**GNN module.**    We consider graph property prediction, i.e., for each graph $\mathcal{G} = (\mathcal{V}, \mathcal{E})$ we have a graph-specific prediction target $y_{\mathcal{G}}$. We suppose that each node $v \in \mathcal{V}$ has an initial feature vector $\boldsymbol{h}_v^0 \in \mathbb{R}^{d_0}$. As `GraphTrans` is a generally-applicable framework that can be used in concert with a

variety of GNNs, we make very few assumptions on the GNN layers that feed into the Transformer subnetwork. A generic GNN layer stack can be expressed as

$$\boldsymbol{h}_v^\ell = f_\ell\left(\boldsymbol{h}_v^{\ell-1}, \{\boldsymbol{h}_u^{\ell-1} | u \in \mathcal{N}(v)\}\right), \quad \ell = 1, \ldots, L_{\text{GNN}} \tag{1}$$

where $L_{\text{GNN}}$ is the total number of GNN layers, $\mathcal{N}(v) \subseteq \mathcal{V}$ is some neighborhood of $v$, and $f_\ell(\cdot)$ is some function parameterized by a neural network. Note that many GNN layers admit edge features, but to avoid notational clutter we omit discussion of them here.

**Transformer module.**   Once we have the final per-node GNN encodings $\boldsymbol{h}_v^{L_{\text{GNN}}}$, we pass these to `GraphTrans`'s Transformer subnetwork. The Transformer subnetwork operates as follows. We first perform a linear projection of the $\boldsymbol{h}_v^{L_{\text{GNN}}}$'s to the Transformer dimension and a Layer Normalization to normalize the embedding:

$$\bar{\boldsymbol{h}}_v^0 = \text{LayerNorm}(\boldsymbol{W}^{\text{Proj}} \boldsymbol{h}_v^{L_{\text{GNN}}}) \tag{2}$$

where $\boldsymbol{W}^{\text{Proj}} \in \mathbb{R}^{d_{\text{TF}} \times d_{L_{\text{GNN}}}}$ is a learnable weight matrix, and $d_{\text{TF}}$ and $d_{L_{\text{GNN}}}$ are the Transformer dimension and the dimension of the final GNN embedding, respectively. The projected node embeddings $\bar{\boldsymbol{h}}_v^0$ are then fed into a standard Transformer layer stack, with no additive positional embeddings, as we expect the GNN to have already encoded the structural information into the node embeddings:

$$a_{v,u}^\ell = (\boldsymbol{W}_\ell^Q \bar{\boldsymbol{h}}_v^{\ell-1})^\top (\boldsymbol{W}_\ell^K \bar{\boldsymbol{h}}_u^{\ell-1})/\sqrt{d_{\text{TF}}} \qquad \alpha_{v,u}^\ell = \underset{w \in \mathcal{V}}{\text{softmax}}(a_{v,w}^\ell)$$
$$\bar{\boldsymbol{h}}_v^{\prime\ell} = \sum_{w \in \mathcal{V}} \alpha_{v,w}^\ell \boldsymbol{W}_\ell^V \bar{\boldsymbol{h}}_w^{\ell-1} \tag{3}$$

where $\boldsymbol{W}_\ell^Q, \boldsymbol{W}_\ell^K, \boldsymbol{W}_\ell^V \in \mathbb{R}^{d_{\text{TF}}/n_{\text{head}} \times d_{\text{TF}}/n_{\text{head}}}$ are the learned query, key, and value matrices, respectively, for a single attention head in layer $\ell$. As is standard, we run $n_{\text{head}}$ parallel attention heads and concatenate the resulting per-head encodings $\bar{\boldsymbol{h}}_v^{\prime\ell}$. Concatenated encodings are then passed to a Transformer fully-connected subnetwork, consisting of the standard Dropout $\to$ Layer Norm $\to$ FC $\to$ nonlinearity $\to$ Dropout $\to$ FC $\to$ Dropout $\to$ Layer Norm sequence, with residual connections from $\bar{\boldsymbol{h}}_v^{\ell-1}$ to after the first dropout, and from before the first fully-connected sublayer to after the dropout immediately following the second fully-connected sublayer.

**<CLS> embedding as a GNN "readout" method.**   As mentioned, for whole-graph classification we require a single embedding vector that describes the whole graph. In the GNN literature, this module that collapses embeddings for every node and/or edge to a single embedding is called the "readout" module, and the most common readout modules are simple mean or max pooling, or a single "virtual node" that is connected to every other node in the network.

In this work, we propose a special-token readout module similar to those used in other applications of Transformers. In text classification tasks with Transformers, a common practice is to append a special <CLS> token to the input sequence before passing it into the network, then to take the output embedding corresponding to this token's position as the representation of the whole sentence. In that way, the Transformer will be trained to aggregate information of the sentence to that embedding, by calculating the one-to-one relationships between the <CLS> token and each other tokens in the sentence with the attention module.

Our application of special-token readout is similar to this. Concretely, when feeding the transformed per-node embeddings $\bar{h}_v^0$, we append an additional learnable embedding $h_{\text{<CLS>}}$ to the sequence, and take the first embedding $\bar{h}_{\text{<CLS>}} \in \mathbb{R}^{d_{\text{TF}}}$ from the transformer output as the representation of the whole graph (note that since we do not include positional encodings, placing the special token at the "beginning" of the sentence has no special computational meaning; the location is chosen by convention). Finally, we apply a linear projection followed by a softmax to generate the prediction:

$$y = \text{softmax}(\boldsymbol{W}^{\text{out}} \bar{\boldsymbol{h}}_{\text{<CLS>}}^{L_{\text{TF}}}). \tag{4}$$

where $L_{\text{TF}}$ is the number of Transformer layers.

This special-token readout mechanism may be viewed as a generalization or a "deep" version of a virtual node readout. While a virtual node method requires every node in the graph to send its

information to the virtual node and does not allow for learning pairwise relationships between graph nodes except within the virtual node's embedding (possibly creating an information bottleneck), a Transformer-style special-token readout method lets the network learn long-range node-to-node relationships in earlier layers before needing to distill them in the later layers.

# 5 Experiments

We evaluate `GraphTrans` on graph classification tasks from three modalities: biology, computer programming, and chemistry. Our `GraphTrans` achieves consistent improvement over all of these benchmarks, indicating the generality and effectiveness of the framework. All of our models are trained with the Adam optimizer [17] with a learning rate of 0.0001, a weight decay of 0.0001, and the default Adam $\beta$ parameters. All Transformer modules used in our experiments have an embedding dimension $d_{\mathrm{TF}}$ of 128 and a hidden dimension of 512 in the feedforward subnetwork. The Transformer baselines described below are trained with only the sequence of node embeddings, discarding the graph structure.

## 5.1 Biological benchmarks

**Datasets.** We choose two commonly used graph classification benchmarks, NCI1 and NCI109 [31]. Each of them contains about 4000 graphs with around 30 nodes on average, representing biochemical compounds. The task is to predict whether a compound contains anti-lung-cancer activity. We follow the settings in [20, 2] for the NCI1 and NCI109, randomly splitting the dataset into training, validation, and test set by a ratio of 8:1:1.

**Training Setup.** We trained `GraphTrans` on both the NCI1 and NCI109 datasets for 100 epochs with a batch size of 256. We run each experiment 20 times with different random seeds and calculate the average and standard deviation of the test accuracies. All the model follows the architecture in Figure 1, with 4 transformer layers and a dropout ratio of 0.1 for both the GNN and Transformer modules. We use two different settings adopted from prior literature for the width and depth of the GNN submodule in `GraphTrans`. The GNN module width and depth in the small `GraphTrans` model are copied from the simple baseline, i.e. the settings in [20], which has a hidden dimension of 128 and 3 GNN layers. The settings of the GNN module in the large `GraphTrans` model are adopted from the default GCN/GIN model provided by OGB, which has a hidden dimension of 300 and 4 GNN layers. We also adopt a cosine annealing schedule [22] for learning rate decay.

**Results.** We report the results on both NCI1 and NCI109 in Table 1. The simple baselines, including GCN Set2Set, SortPool, and SAGPool, are taken from [20], while the strong baselines [13], as well as the FA layer [2]. In Table 1, Our Graph Transformer (small) has the same architecture as the simple baseline but improves the average accuracy by 7.1% for NCI1 and 5.1% for NCI109. We also tested the framework with GIN as the encoder (`GraphTrans` (large)) to align with the settings in the strong baseline, which also significantly improves the accuracy of the strong baseline by 1.1% for NCI1 and the 8.2% for NCI109, even without the deep GNN, using 4 layers instead of 8.

## 5.2 Chemical benchmarks

**Datasets.** For chemical benchmarks, we evaluate our `GraphTrans` on a dataset larger than NCI dataset, `molpcba` from the Open Graph Benchmark (OGB) [15]. It contains 437929 graphs with 28 nodes on average. Each graph in the dataset represents a molecule, where nodes and edges are atoms and chemical bonds, respectively. The task is to predict the multiple properties of a molecule. We use the standard splitting from the benchmark. The performance on the GIN and GIN-Virtual baselines are as reported on the OGB leaderboard [15].

**Training Setups.** All the GNN modules in the experiments follow the settings of the default GIN model provided in OGB, with 4 layers and 300 hidden dimension. We train all the models for 100 epochs with a batch size of 256 and report the test result with the best validation ROC-AUC. For both GNN and Transformer modules, we apply a dropout of 0.3. We use GIN as the baseline and the GNN module, since it performs better than GCN models on the Molpcba dataset.

Table 1: **NCI biological datasets**  `GraphTrans` outperforms past baselines on both NCI1 and NCI109 test accuracy while using fewer GNN layers than prior SOTA baselines.

| Model | GNN Type | GNN layer count | NCI1 (%) | NCI109 (%) |
|---|---|---|---|---|
| Set2Set [29, 20] | GCN | 3 | 68.6± 1.9 | 69.8±1.2 |
| SortPool [39, 20] | GCN | 3 | 73.8±1.0 | 74.0±1.2 |
| SAGPool$_h$ [20] | GCN | 3 | 67.5±1.1 | 67.9±1.4 |
| SAGPool$_g$ [20] | GCN | 3 | 74.2±1.2 | 74.1±0.8 |
| Errica et al. [13] | GIN | 8 | 80.0±1.4 | – |
| Alon and Yahav [2] | GIN | 8 | 81.5±1.2 | – |
| Transformer [28] | – | – | 68.5±2.6 | 70.1± 2.3 |
| `GraphTrans` (small) | GCN | 3 | **81.3±1.9** | **79.2±2.2** |
| `GraphTrans` (large) | GIN | 4 | **82.6±1.2** | **82.3±2.6** |

Table 2: **OpenGraphBenchmark Molpcba dataset**  Overall, `GraphTrans` outperforms competitive baselines with two backbone GNN architectures.

| Model | Valid ROC-AUC | Test ROC-AUC |
|---|---|---|
| GCN [18] | 0.2059±0.0033 | 0.2020±0.0024 |
| GIN [36] | 0.2305±0.0027 | 0.2266±0.0028 |
| GCN-Virtual [18] | 0.2495±0.0042 | 0.2424±0.0034 |
| GIN-Virtual [36] | 0.2798±0.0025 | 0.2703±0.0023 |
| Transformer [28] | 0.1316±0.0012 | 0.1281±0.0039 |
| `GraphTrans` (GIN) | **0.2893±0.0050** | 0.2756±0.0039 |
| `GraphTrans` (GIN-Virtual) | 0.2867±0.0022 | **0.2761±0.0029** |

**Results.**  In Table 2, we report the ROC-AUC on validation and test set of Molpcba. Though Transformer alone works very badly on this dataset, our `GraphTrans` still improves the ROC-AUC of the GIN and GIN-Virtual baseline. It indicates that our design could take benefit from both the local graph structure learned by the GNN and the long-range concept retrieved by the Transformer module based on the GNN embeddings.

### 5.3  Computer programming benchmark

**Datasets.**  For the computer programming benchmark, we also adopt a large dataset, code2 from OGB, which has 45741 graphs each with 125 nodes on average. The dataset is a collection of Abstract Syntax Trees (ASTs) from about 450k Python method definitions. The task is to predict the sub-tokens forming the method name, given the method body represented by the AST. We also adopt the standard dataset splitting from the benchmark. All baseline performances are as reported on the OGB leaderboard.

**Training Setups.**  We also apply the default settings of GCN for Code2 from OGB, with 4 GNN layers, 300 hidden dimension, and a dropout ratio of 0.0. We apply a dropout ratio of 0.3 to the Transformer module to avoid overfitting. We train all the models for 30 epochs with a batch size of 16, due to the large scale of the dataset. For the `GraphTrans` (PNA) model, we follow the settings in [26], with a hidden embedding of 272 for the GNN module and a weight decay of 3e-6. The only difference is that we still use the learning rate of 0.0001, instead of the heavily tuned 0.00063096 [26]. We run each experiment 5 times and take the average and standard deviation of the F1 score.

**Results.**  In Table 3, we compare our `GraphTrans` with top tier architectures on the leaderboard on Code2 dataset. As the average number of nodes in each graph increases, the global information becomes more important as it becomes more difficult for the GNN to gather information from

Table 3: **OpenGraphBenchmark Code2 dataset** All the baselines are collected from the OGB leaderboard. `GraphTrans` outperforms the state-of-the-art DAGNN. The improvement based on PNA model indicates that our method is orthogonal to the type of GNN module.

| Model | Valid F1 score | Test F1 score |
|---|---|---|
| GIN [36] | 0.1376±0.0016 | 0.1495±0.0023 |
| GCN [18] | 0.1399±0.0017 | 0.1507±0.0018 |
| GIN-Virtual [36] | 0.1439±0.0026 | 0.1581±0.0020 |
| GCN-Virtual [18] | 0.1461±0.0013 | 0.1595±0.0018 |
| PNA [8] | 0.1453±0.0025 | 0.1570±0.0032 |
| DAGNN (SOTA) [27] | 0.1607±0.0040 | 0.1751±0.0049 |
| Transformer [28] | 0.1546±0.0018 | 0.1670±0.0015 |
| GraphTrans (GCN) | 0.1599±0.0009 | 0.1751±0.0015 |
| GraphTrans (PNA) | **0.1622±0.0025** | **0.1765±0.0033** |
| GraphTrans (GCN-Virtual) | **0.1661±0.0012** | **0.1830±0.0024** |

Table 4: **Ablation of Transformer module** On the Code2 dataset, only training the Transformer module in `GraphTrans` with a frozen pre-trained GNN module also improves the F1-score. It indicates that training the Transformer on GNN embeddings can learn information that is not captured by the GNN.

| Model | Valid F1 score | Test F1 score |
|---|---|---|
| Pre-trained GCN-Virtual | 0.1457 | 0.1574 |
| GraphTrans, pre-trained GCN-Virtual, frozen GNN | 0.1479 | 0.1616 |
| GraphTrans, pre-trained GCN-Virtual, fine-tuned GNN | **0.1564** | **0.1733** |

nodes far away. Even without heavy tuning, `GraphTrans` significantly outperforms the state-of-the-art (DAGNN) [27] on the leaderboard. We also include the results for the PNA model and our `GraphTrans` with the PNA model as the GNN encoder. Our `GraphTrans` also significantly improves the result, which indicates that our architecture is orthogonal to the variants of the GNN encoder module.

## 5.4 Transformers can capture long-range relationships

As we previously observed in Figure 2 and discussed in Section 3, the attention inside the transformer module can capture long-range information that is hard to be learned by the GNN module.

To further verify the hypothesis, we designed an experiment to show that the Transformer module can learn additional information to the GNN module. In Table 4, we first pretrain a GNN (GCN-Virtual) until converge on the Code2 dataset, and then freeze the GNN model and plug our Transformer module after it. By training the model on the training set with a fixed GNN module, we can still observe a 0.0022 F1-score improvement on validation set and 0.0042 on test set. It indicates that the Transformer can learn additional information that is hard to be learned by the GNN module along.

With pretrained and unfrozen GNN module, our `GraphTrans` can achieve an even higher F1-score. That may because the GNN module can now focus on learning the local structure information, by leaving the long-range information learning to the Transformer layer after it. The model benefits from the specialization as mentioned in [34]. Note that for all the experiments in Table 4, we do not concatenate the embeddings from the input graph to the input of Transformer for simplicity.

## 5.5 Effectiveness of <CLS> embedding

In Figure 2b, we can observe that row 18 (the last row is for <CLS>) has dark red on multiple columns, which indicates that the <CLS> learns to attend to important nodes in the graph to learn the representation for the whole graph.

Table 5: **Ablation of `<CLS>` token** The `mean` and `last` are two commonly used embedding aggregation method for sequence classification.

| Model | Valid | Test |
|---|---|---|
| GraphTrans, mean | 0.1398 | 0.1509 |
| GraphTrans, last | 0.1566 | 0.1716 |
| GraphTrans, <CLS> | 0.1593 | 0.1784 |
| GraphTrans, <CLS>, cat | **0.1670** | **0.1810** |

Table 6: **Scalability of Transformer to large graphs** We profile our Code2 model on random graphs and list runtime in milliseconds. `GraphTrans` scales comparably to the GCN model due to the high cost neighbor sampling in GNN training. With graphs over 1000 nodes, `GraphTrans` is no less scalable than the GCN baseline.

| Node count | Model | Edge Density | | | |
|---|---|---|---|---|---|
| | | 20% | 40% | 60% | 80% |
| 500 | GCN-Virtual [18] | **44.3** | 58.5 | 79.3 | 99.0 |
| | GraphTrans (GCN) | 48.4 | **57.5** | **76.4** | **93.7** |
| 1000 | GCN-Virtual [18] | 99.1 | 171.8 | 249.5 | OOM |
| | GraphTrans (GCN) | **96.9** | **168.4** | **244.3** | OOM |
| 1200 | GCN-Virtual [18] | 131.8 | 237.7 | **OOM** | **OOM** |
| | GraphTrans (GCN) | **127.9** | **236.6** | **OOM** | **OOM** |

We also examined the effectiveness of our `<CLS>` embedding quantitatively. In Table 5, we tested several common methods to for sequence classification. The `mean` operation averages the output embeddings of the transformer to a single graph embedding; the `last` operation takes the last embedding in the output sequence as the graph embedding. The quantitative results indicate that the `<CLS>` embedding is most effective with 0.0275 improvements on the test set, as the model can learn to retrieve information from different nodes and aggregate them into one embedding. The concatenation of the embeddings in the input graph and the input embeddings of the transformer can further improve the validation and test F1-score to 0.1670 and 0.1733.

## 5.6 Scalability

To quantitatively benchmark how `GraphTrans` scales with large graphs over 100 nodes, we ran a microbenchmark of iteration time for training with varying graph size and edge density. We train baselines on randomly generated Erdos-Renyi graphs with a varying number of nodes and edge density. As shown in the Table 6, our `GraphTrans` model scales at least as well as the GCN model when the number of nodes and edge density increases. Both GCN and `GraphTrans` see out of memory errors (OOM) with large dense graphs, but we note that `GraphTrans` had similar memory consumption to the GCN baseline.

## 5.7 Computational efficiency

To evaluate the overhead that our `GraphTrans` adds over a specific GNN backbone, we evaluate the forward pass runtime and backward pass runtime per iteration. We normalize models to have roughly similar parameter counts. The results are shown in Table 7. For the NCI1 dataset, `GraphTrans` is actually faster to train than a comparable GCN model. For the OGB-molpcba and OGB-Code2 datasets, `GraphTrans` is 7-11% slower than the baseline GNN architectures.

## 5.8 Number of parameters

We compare the number of parameters of the GNN baseline and the `GraphTrans` on different dataset in Table 8. Overall, `GraphTrans` only increases total parameters marginally for Molpcba and NCI.

Table 7: **Speedup of Transformer module** For `GraphTrans` models trained over the NCI1, OGBG-Molpcba and OGBG-Code2 datasets, we find that the Transformer module adds minimal overhead. Speedup is the training iteration speed compared to GNN based model; larger number indicates a faster running speed.

| Dataset | Method | Forward time (ms) | Backward time (ms) | Speedup |
|---------|--------|-------------------|--------------------|---------|
| NCI1 | GCN-Virtual [18] | 22.27 ± 2.04 | 14.35 ± 2.46 | 1.00× |
| | Transformer | 12.31 ± 1.68 | 9.32 ± 1.68 | 1.69× |
| | GraphTrans | 15.01 ± 1.63 | 12.04 ± 2.25 | 1.35× |
| Molpcba | GCN-Virtual [18] | 14.79 ± 2.54 | 12.75 ± 3.00 | 1.00× |
| | Transformer | 12.34 ± 1.43 | 10.52 ± 1.60 | 1.20× |
| | GraphTrans | 16.55 ± 2.93 | 14.3 ± 3.15 | 0.89× |
| Code2 | GCN-Virtual [18] | 22.97 ± 6.13 | 38.53 ± 6.92 | 1.00× |
| | Transformer | 31.01 ± 9.00 | 33.30 ± 16.09 | 0.96× |
| | GraphTrans | 34.93 ± 6.85 | 31.14 ± 12.90 | 0.93× |

Table 8: **Parameter count** Overall, `GraphTrans` achieves improved accuracy with a minor increase in parameters.

| Dataset | GNN | GraphTrans | Delta |
|---------|-----|------------|-------|
| Molpcba | 3.4M | 4.2M | 0.8M |
| NCI | 0.4M | 0.5M | 0.1M |
| Code2 | 12.5M | 9.1M | -3.4M |

For Code2, `GraphTrans` is substantially more parameter-efficient than the GNN while improving test F1 score from 0.1629 to 0.1810. One reason for improved parameter efficiency is that the Transformer reduces feature dimension before the expensive final prediction layer.

# 6 Conclusion

We proposed `GraphTrans`, a simple yet powerful framework for learning long-range relationships with GNNs. Leveraging recent results that suggest structural priors may be unnecessary or even counterproductive for high-level, long-range relationships, we augment standard GNN layer stacks with a subsequent permutation-invariant Transformer module. The Transformer module acts as a novel GNN "readout" module, simultaneously allowing the learning of pairwise interactions between graph nodes and summarizing them into a special token's embedding as is done in common NLP applications of Transformers. This simple framework leads to surprising improvements upon the state of the art in several graph classification tasks across program analysis, molecules and protein association networks. In some cases, `GraphTrans` outperforms methods that attempt to encode domain-specific structural information. Overall, `GraphTrans` presents a simple yet general approach to improve long-range graph classification; next directions include applications to node and edge classification tasks as well as further scalability improvements of the Transformer to large graphs.

# 7 Acknowledgements

We thank Ethan Mehta, Azade Nazi, Daniel Rothschild, Adnan Sherif and Justin Wong for thoughtful discussions and feedback. In addition to NSF CISE Expeditions Award CCF-1730628, this research is supported by gifts from Amazon Web Services, Ant Group, Ericsson, Facebook, Futurewei, Google, Intel, Microsoft, Scotiabank, and VMware.

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
