# OpenReview forum: "Representing Long-Range Context for Graph Neural Networks with Global Attention"
_NeurIPS.cc/2021/Conference — NeurIPS 2021 Poster_

### Official Review · Reviewer_SioY · 2021-07-13

**Rating:** 5
**Confidence:** 4

**Summary:**

Current GNN methods fail at utilizing long-range dependencies in graphs due to over smoothing. However, in some cases, long-range information might be useful. Inspired by the success of transformers in computer vision tasks, this paper tackles the challenge of learning long-range dependencies in GNNs by simply augmenting a common GNN architecture with a transformer sub-network. They also borrow a “readout” token from NLP which improves the final learned representation of the graph.

**Limitations And Societal Impact:**

The authors shortly discuss the limitations in the checklist section as mentioned above in the main review.

**Main Review:**

1. **Scalability - $O(n^2)$ memory complexity**

   My main concern regarding this paper - is its scalability which is very shortly discussed in the checklist. The transformer subnetwork has to compute and hold in memory attention maps of size, which for graphs already reaching ~100 nodes it could raise a memory issue. Unlike NLP tasks where a usual sentence would contain at most tens of words, graph learning tasks could reach graphs of thousands of nodes.

2. **Originality**

   The authors fail to mention that global attention in graph networks has already been introduced in [1]. The difference between [1] and these works is that the global attention layers are augmented to each GNN layer in [1] and not just at the end. Also, [1] deals with the scalability issue by learning a low-rank representation of the attention matrix having a memory complexity of $O(kn)$ where $k$ is the rank and $k<<n$.

   I therefore feel that the differences between the methods are quite minor unless we see empirical evidence or theoretical justification as to why this work is better and novel than [1].

3. **Experimental Evaluation**

   - The method is only tested on graph classification tasks. Is the transformer subnetwork also beneficial for node classification and link prediction?
   - Significantly better performance has already been achieved on the NC1 and NC109 datasets in [2]. Is the setting different?




***Other comments***

- Typo in the title of section 3

  > Motivation: Learning to model long-range pairwise interactions in gr

  What is gr?



---

[1] Puny, O., Ben-Hamu, H., & Lipman, Y. (2020). Global Attention Improves Graph Networks Generalization. *arXiv preprint arXiv:2006.07846*.

[2] Morris, Christopher et al. “Weisfeiler and Leman go sparse: Towards scalable higher-order graph embeddings.” arXiv: Data Structures and Algorithms (2020): n. pag.


---
After reading the authors' responses and the other reviews I think the paper still lacks novelty, however it shows strong performance. I therefore update my score.

**Time Spent Reviewing:**

5

---

> ### Author Response · Authors · 2021-08-10
> **Thank you for your feedback Reviewer SioY**
>
> Thank you for your careful and thoughtful review. We respond to your key criticisms below. We also would like to refer you to the shared response where we discuss the parameter efficiency of GraphTrans.
>
> ### Comparison with Puny et al. 2020:
> Thank you for the reference -- we missed this submission as it was under review.
>
> Using the open-source release of LRGA by the authors, we evaluated LRGA on the NCI biological datasets. We found that LRGA underperformed GraphTrans:
>
> |                               |   Val. Acc.   |   Test Acc.   |
> |------------------------------:|:-------------:|:-------------:|
> |      Baseline (GCN + virtual) | 68.71 ± 4.12% | 67.20 ± 2.17% |
> |       LRGA (Puny et al. 2020) | 68.13 ± 2.82% | 66.86 ± 1.33% |
> |            Transformer w/ GNN | 73.83 ± 1.77% | 72.18 ± 2.31% |
> | GraphTrans (ours, replicated) | 84.29 ± 1.74% | 83.00 ± 1.44% |
>
> Notably, LRGA quickly overfits to the training set. The validation accuracy of LRGA increases in the first 30 epochs and starts to drop after that, while the validation accuracy of other methods can continuously increase. Even with early-stopping based on validation metrics, LRGA performs 5.7% and 5.3% worse than GraphTrans on validation and test sets respectively. That implies that LRGA does not have enough capacity to learn the global information for the graph classification task.
>
> ### Scalability of self-attention over large graphs
> Our GraphTrans scales well on the current graph classification benchmarks, where Code2 has large graphs up to 10k nodes (Code2 graphs have 125 nodes on average but its largest graph has 36,123 nodes). As the hidden dimension of the representation is small, we have not found the Transformer to scale poorly with large graphs. We have not found common graph classification tasks with more than 10k nodes. However, we acknowledge that direct application of GraphTrans to graphs larger than 10k+ nodes will be challenging.
>
> We are also investigating the scalability of our GraphTrans model. The computation and memory cost of the model is very task specific. For example, the GNN model works well on sparse graphs, but for more dense graphs, the cost will significantly increase, while the Transformer still remains the same. Also, for the complexity of the Transformer model, it has been widely researched in the NLP area. Some efficient versions of the transformer are out of the box for the scalability problem, e.g. the LiteTransformer[1] with less FLOPs, Reformer[2] with $O(L \log L)$ complexity, and Performer[3] with both less computation and memory complexity.
>
> [1] Wu, Zhanghao, et al. "Lite transformer with long-short range attention." ICLR 2020.
> [2] Kitaev, Nikita, Łukasz Kaiser, and Anselm Levskaya. "Reformer: The efficient transformer." ICLR2020.
> [3] Choromanski, Krzysztof, et al. "Rethinking attention with performers." ICLR 2021.
>
> ### WL NCI1 and NC109 baseline
> The “Weisfeiler and Leman go sparse” is not a neural baseline, which is out of the scope of what we are discussing. Our strong baseline follows the neural baseline proposed in Errica et al. “A fair comparison of graph neural network” which was accepted by the ICLR 2020, and the latest method FALayer proposed in Alon et al. “On the Bottleneck of Graph Neural Networks and its Practical Implications”, which was accepted by the ICLR 2021.
>
> ### Clarifying title of section 3
> The correct title should be “Motivation: Learning to model long-range pairwise interactions on graphs”. Our apologies for this typesetting issue.

---

> ### Author Response · Authors · 2021-08-12
> **New experiment: scalability of GraphTrans to large graphs**
>
> We are excited that we are able to present quantitative benchmarks that hopefully will answer your questions regarding the scalability of GraphTrans to large graphs. Note that all experiments are performed with a batch size of 2 (to avoid bs=1 anomalies in PyTorch) on a 16GB V100 GPU (p3.2xlarge instance).
>
> ### Microbenchmark: scalability of GraphTrans for large graphs
> To quantitatively benchmark how GraphTrans scales with large graphs over 100 nodes, we ran a microbenchmark w/ varying graph size and edge density. We train baselines on randomly generated Erdos-Renyi graphs with a varying number of nodes and edge density.
>
> As shown in the table below, our GraphTrans model scales at least as well as the GCN model when the number of nodes and edge density increases. Both GCN and GraphTrans see out of memory errors (OOM) with large dense graphs, but we note that GraphTrans had similar memory consumption to the GCN baseline.
>
> **Number of Nodes: 500**
>
> | Model       | Edge Density  |  20% |  40% |  60% |  80% |
> |------------:|:---------------|:-----:|:-----:|:-----:|:-----:|
> | GCN-Virtual | Forward Time (ms)  | 19.3 | 26.5 | 36.7 | 47.7 |
> |             | Backward Time (ms) | 25.0 | 32.0 | 42.6 | 51.3 |
> |             | Total Time (ms)    | **44.3** | 58.5 | 79.3 | 99.0 |
> |             |               |      |      |      |      |
> | GraphTrans  | Forward Time (ms)  | 22.8 | 29.9 | 40.2 | 49.8 |
> |             | Backward Time (ms) | 25.6 | 27.6 | 36.2 | 43.9 |
> |             | Total Time (ms)   | 48.4 | **57.5** | **76.4** | **93.7** |
>
>
> **Number of Nodes: 1000**
>
> | Model       | Edge Density  |   20% |    40% |    60% | 80% |
> |------------:|:---------------|:-----:|:-----:|:-----:|:-----:|
> | GCN-Virtual | Forward Time (ms)  |  50.5 |  91.18 | 136.66 | OOM |
> |             | Backward Time (ms) | 48.57 |  80.64 | 112.82 | OOM |
> |             | Total Time (ms)    | 99.07 | 171.82 | 249.48 | OOM |
> |             |               |       |        |        |     |
> | GraphTrans  | Forward Time (ms)  | 52.64 |  91.92 | 136.34 | OOM |
> |             | Backward Time (ms) | 44.29 |  76.51 | 107.91 | OOM |
> |             | Total Time (ms)    | **96.93** | **168.43** | **244.25** | OOM |
>
>
> **Number of Nodes: 1200**
>
> | Model       | Edge Density  |    20% |    40% | 60% | 80% |
> |------------:|:---------------|:-----:|:-----:|:-----:|:-----:|
> | GCN-Virtual | Forward Time (ms)  |  69.47 | 129.05 | OOM | OOM |
> |             | Backward Time (ms) |  62.34 | 108.69 | OOM | OOM |
> |             | Total Time (ms)    | 131.81 | 237.74 | OOM | OOM |
> |             |                    |        |        |     |     |
> | GraphTrans  | Forward Time (ms)  |  70.37 | 132.47 | OOM | OOM |
> |             | Backward Time (ms) |  57.51 | 104.09 | OOM | OOM |
> |             | Total Time (ms)    | **127.88** | **236.56** | OOM | OOM |

---

> ### Comment · Area_Chair_R7NU · 2021-08-29
> **Please give us your reaction to the author response**
>
> Thanks.
> The AC

---

### Official Review · Reviewer_5JC1 · 2021-07-15

**Rating:** 6
**Confidence:** 5

**Summary:**

This paper proposes to combine GNN with a permutation-invariant transformer called GraphTrans for the graph classification tasks. The GNN module is used to learn representations from graph structure,  while the transformer module without a positional encoding is to learn long-range pairwise relationships, with "CLS"-like readout to obtain a global graph embedding. Experimental results show that the proposed simple architecture leads to state-of-the-art results on several graph classification tasks, better than other GNN models. These results also suggest that modeling all pairwise node-node interactions in the graph is
particularly important for large graph classification tasks.

**Limitations And Societal Impact:**

See above comments.

**Main Review:**

Pros:
1. GraphTrans proposes to use transformer layers (without position embeddings) to learn the pairwise node interactions and summarize the graph representation after a normal GNN model.
2. Adding this kind of transformer to the GNN improves the performance of various graph classification tasks.

Cons:
1. The number of parameters in GraphTrans is much bigger (maybe doubled) than its GNN version without a transformer (I assume all experiments used 4 layers of transformers as stated on page 6). Thus the improvement presented in Table1-3 may not be fair comparisons.
2. Combining Transformer with GNN is very common for NLP tasks (pre-trained transformer LM as an encoder for input text sequence), followed by GNN on graphs such as knowledge graphs or dependency parsing trees. Thus the novelty of GraphTrans in this paper is rather limited. I do not agree with the statement that the transformer in GraphTrans is learning "long-range" relationships, because the module is computing any pair of feature interactions, with attention weights to control the contribution of each node to the final graph embedding. So it would make sense to try another set of experiments: to switch the order of transformer and GNN, i.e use transformer module first and then GNN, and see the results.
3. Since the transformer uses self-attention, it would make sense to compare to GAT-like GNN models (for example, the recent paper on direct multi-hop attention network, although this paper only showed results on node classification and knowledge graph embedding, not on graph classification.
4. The limitation of GraphTrans is that it is only for graph-level classification, and is not suited to node classification.

Questions:
1. The last row of ablation results in Table 5 is explained in lines 290-292: not sure how the embeddings of input graph are concatenated with input embedding of the transformer, and where they are used.

Some typos on line 110, Figure 2 caption, line 286, line 292 need to be fixed.

***
After reading the authors' responses to my questions, I have adjusted my review score. I am not convinced on the novelty side of the method. However, if only considering the graph classification task, their experimental results have shown the effectiveness of the proposed method.

**Time Spent Reviewing:**

6

---

> ### Author Response · Authors · 2021-08-10
> **Thank you for your feedback Reviewer 5JC1**
>
> Thank you for your time and your careful review. We are responding to your key concerns now and will follow up with any further experimental results (time permitting) in a second reply.
>
> ### Parameter count of GraphTrans versus GNN baselines
> We calculated the number of parameters for GraphTrans and a GNN baseline in the General Response (2, Cost of our method, parameter efficiency). Overall, GraphTrans has relatively low overhead when compared with a baseline GNN. For the Code2 dataset, our method is significantly more parameter efficient.
>
> ### Evaluating alternative architecture: Transformer then GNN
> This is a great idea inspired by a common pattern from NLP. We evaluated your proposed architecture (with GNN after Transformer) on both the NCI1, NCI109 and OGBG-code2. We can observe that without the graph structure encoded by the GNN module, the model cannot fully utilize the capacity of the Transformer for graph classification tasks.
>
> **Transformer-GNN results on NCI1 benchmark**
>
> |                         | Validation accuracy   | Test accuracy   |
> |------------------------:|:-----------:|:-----------:|
> | Baseline (SAGPool$_g$)  | (not reported) | 74.2 ± 1.2% |
> | Transformer-GNN (small) | 73.5 ± 1.7% | 72.1 ± 2.6% |
> | Transformer-GNN (large) | 78.8 ± 2.2% | 76.6 ± 1.7% |
> | GraphTrans (small)      | 81.9 ± 1.7% | 80.2 ± 1.9% |
> | GraphTrans (large)      | 84.4 ± 1.6% | 83.0 ± 1.6% |
>
> **Transformer-GNN results on NCI109 benchmark**
>
> |                         | Validation accuracy   | Test accuracy   |
> |------------------------:|:-----------:|:-----------:|
> | Baseline (SAGPool$_g$)  | (not reported) | 74.1 ± 0.8% |
> | Transformer-GNN (small) | 73.2 ± 2.1% | 71.4 ± 1.8% |
> | Transformer-GNN (large) | 77.8 ± 1.8% | 75.4 ± 1.8% |
> | GraphTrans (small)      | 79.8 ± 2.0% | 79.0 ± 2.5% |
> | GraphTrans (large)      | 84.6 ± 2.0% | 82.5 ± 2.0% |
>
> **Transformer-GNN results on OGB-Code2 benchmark**
>
> |                          | Validation F1 score | Test F1 score |
> |-------------------------:|:-------:|:-------:|
> | Baseline (GCN + virtual) | 0.1461  | 0.1595  |
> |          Transformer-GNN | 0.1475  | 0.1578  |
> |               GraphTrans | 0.1670  | 0.1810  |
>
> ### Comparison with Graph Attention Networks (GAT)
> GAT as originally proposed does not perform graph classification and therefore is not currently benchmarked on the OGB leaderboard nor the NCI datasets. Extending GAT from node classification to graph classification is an open area of research.
>
> GAT utilizes attention over local graph neighborhoods, not global, so we do not expect it to improve modelling long-range dependencies. Our framework is also orthogonal to the choice of backbone GNN architecture. We finally note that we outperform Self-attention Graph Pooling which is a method that utilizes global attention.
>
> ### Method tested on graph classification, not node classification
> In our paper, we mainly focus on the graph classification task as it can benefit the most from modelling long-range dependencies in the graph. Moreover, graphs found in graph classification tasks tend to be more dense than those in node or edge classification; therefore global attention is more relevant.
>
> It is also possible to adopt our methods for node classification tasks by replacing the graph pooling method to a node-wise projection and the graph-based loss to node-based. We will leave it as a future work to corporate the global information to node classification tasks.

---

### Official Review · Reviewer_XZ8p · 2021-07-16

**Rating:** 7
**Confidence:** 2

**Summary:**

This work proposes to use a transformer that attends over all nodes in the graph, given an intermediate representation obtained using a standard GNN module. Contributions 1) usage of a self-attention mechanism that takes into account all nodes in the graph, as opposed to previous methods discussed in the related work. 2) the removal of the positioning encoding to allow permutation invariance. 3) Quantitative results on several datasets, overcoming the baselines and supporting the method with empirical evidence.


**Limitations And Societal Impact:**

The limitations are not discussed.

**Main Review:**

The paper is well written and clear. The work is well positioned with respect to existing work, to the best of my knowledge, clearly stating what are the benefits and contributions of the proposed method.

The usage of a transformer to learn long-range dependencies is well motivated, drawing evidence from the NLP and vision communities. The method is simple and flexible to the GNN backbone used, as shown in the experimental section.

The technical novelty is moderate, but the method has been tested on several datasets, showing good results and overcoming a basic Transformer and other GNNs.


Additional questions/comments:
- In equation 3, it seems it should be $ \alpha^l_{v,w} = softmax(\alpha^l_{v,u}) $
- What happens if a positional encoding is used? I would expect the quantitative results to degrade, but it might be a relevant study to show that the choice of no positional encoding is beneficial for the model.
- What is the computational cost overhead of adding GraphTrans to a specific backbone? And how many more parameters does it introduce?

Overall, it is a relevant contribution to the field that comes from a sound motivation and is sufficiently benchmarked in the experimental section.


**Time Spent Reviewing:**

2

---

> ### Author Response · Authors · 2021-08-11
> **Thank you for your feedback, Reviewer XZ8p**
>
> Thank you for taking the time to review our draft and for your helpful feedback. We appreciate that you found our draft clear and well motivated. We are profiling GraphTrans to evaluate its computational efficiency and hope to have more quantitative results shortly.
>
> ### Transformer with positional encoding
> Your question regarding an ablation with positional encoding was also related by Reviewer ooyr. Please refer to the General Response (1, Evaluating positional encodings for Transformer). We provide some new experiments in that response.
>
> ### Cost of our method (parameter efficiency)
> We calculated the number of parameters for GraphTrans and a GNN baseline in the General Response (2). Overall, GraphTrans remains fairly efficient with little impact on total parameter count. For the Code2 dataset, GraphTrans is considerably more parameter efficient than the GNN baseline.

---

> ### Author Response · Authors · 2021-08-12
> **New experiment: evaluating computational overhead of GraphTrans versus GNN backbone**
>
> We are excited that we are able to present quantitative benchmarks that hopefully will answer your questions regarding the computational overhead of GraphTrans. Note that all experiments are performed with a batch size of 2 (to avoid bs=1 anomalies in PyTorch) on a 16GB V100 GPU (p3.2xlarge instance).
>
> ### Macrobenchmark: Overhead of GraphTrans over GNN backbone
> To evaluate the overhead that GraphTrans adds over a specific GNN backbone, we evaluate the forward pass runtime and backward pass runtime per iteration. Per-iteration runtimes are averaged across at least 1000 iterations per trial, and we report mean runtime as well as standard deviation. We normalize models to have roughly similar parameter counts. Speedups are reported (where a speedup >1 means that GraphTrans was faster than the GNN and a speedup <1 means that GraphTrans was slower than the GNN).
>
> For the NCI1 dataset, GraphTrans is actually faster to train than a comparable GCN model. For the OGB molpcba and OGB code2 datasets, GraphTrans is 7-11% slower than the baseline GNN architectures. This cost is relatively minor given the substantial increase in accuracy that GraphTrans achieves.
>
> This result may be surprising, but we would like to note that our implementation of the Transformer can be highly optimized on GPUs (for example, via opt_einsum). We also find that our Transformer operator operates at a higher arithmetic intensity than the tested GNN layers; therefore, GraphTrans achieves higher GPU utilization than GNN baselines.
>
> **Computational cost evaluation for NCI1 dataset:**
>
> |              Method | Forward time (ms) | Backward time (ms) | Training iteration speedup over GNN |
> |--------------------:|:-----------------:|:------------------:|:---------------------------:|
> | GCN w/ virtual node |    22.27 ± 2.04   |    14.35 ± 2.46    |             1.00            |
> |         Transformer |   12.31 ± 1.68    |    9.32 ± 1.68     |             1.69            |
> |   GraphTrans (ours) |   15.01 ± 1.63    |    12.04 ± 2.25    |             1.35            |
>
> **Computational cost evaluation for molpcba dataset:**
>
> |              Method | Forward time (ms) | Backward time (ms) | Training iteration speedup over GNN |
> |--------------------:|:-----------------:|:------------------:|:---------------------------:|
> | GIN w/ virtual node |    14.79 ± 2.54   |      12.75 ± 3     |             1.00            |
> |         Transformer |    12.34 ± 1.43   |     10.52 ± 1.6    |             1.20            |
> |   GraphTrans (ours) |    16.55 ± 2.93   |     14.3 ± 3.15    |             0.89            |
>
> **Computational cost evaluation for code2 dataset:**
>
> |              Method | Forward time (ms) | Backward time (ms) | Training iteration speedup over GNN |
> |--------------------:|:-----------------:|:------------------:|:---------------------------:|
> | GCN w/ virtual node |    22.97 ± 6.13   |    38.53 ± 6.92    |              1.00              |
> |         Transformer |     31.01 ± 9.00              |     33.30 ± 16.09               |     0.96                        |
> |   GraphTrans (ours) |    34.93 ± 6.85   |    31.14 ± 12.9    |             0.93            |

---

### Official Review · Reviewer_ooyr · 2021-07-16

**Rating:** 7
**Confidence:** 4

**Summary:**

The paper proposes that adding transformers on top of Graph Neural Networks layers aids performance, and furthermore that adding a readout position is a better way to get a final output vs. simple aggregation strategies.  Other papers have in various forms attempted to add transformers to neural networks, but this paper uses a position-encoding free, approach that performs better while being simpler.  This also demonstrates the importance of long range information propagation in solving GNN problems.  They demonstrate the superior performance of their method empirically in a variety of interesting benchmarks.


**Ethical Concerns:**

No ethical concerns

**Limitations And Societal Impact:**

They have address limitations and potential negative societal impacts.

**Main Review:**

Originality: Other papers have applied Transformers on top of GNNs but this paper demonstrates that a simpler approach which completely ignores position encoding for transformer part actually results in better performance in the tested applications.  It also introduces using the <CLS> readout position, as done in sequence applications, helps performance in the graph neural network application, which is apparently novel in this context as well.

Impact: The authors empirically demonstrate performance improvements on a variety of datasets.  Furthermore, this paper can also lead that to many other potential network architectures that can further improve performance, such as alternating between transformer and GNN layers.  It also shows the importance of long distance information propagation in solving real graph problems.

Clarity:  The paper is well written and easy to follow.  Diagrams are of a good quality and results and their exposition are easy to follow.

Quality:  Submission appears technically sound and is supported by empirical , if not theoretical, evidence.  Ablation studies were performed to narrow down the contributions by the various contributions suggested by the paper, showing the importance of each stage.

As a comment however, the paper doesn't demonstrate that actual position encoding isn't helpful - instead it shows that the currently tested schemes are not helpful.  There might be other encodings that perform even better than the suggested approach.


**Time Spent Reviewing:**

4

---

> ### Author Response · Authors · 2021-08-10
> **Thank you for your feedback Reviewer ooyr**
>
> Thank you for your careful and thoughtful questions and suggestions. We are encouraged that you found our draft to be clear and technically sound.
>
> ### Transformer positional encodings
> Please refer to the General Response (1, Evaluating positional encodings for Transformer) for experimental results with sin-cos encodings. We agree that an alternative positional encoding method could exist and outperform GraphTrans. Specifically, it would be great if a positional encoding would enable removing the GNN from GraphTrans. Hopefully the simple encoding-free approach we present in GraphTrans (with open and reproducible artifacts) can enable future work to discover such an encoding.

---

### Author Response · Authors · 2021-08-10
**Author response to common reviewer concerns**

We sincerely thank the reviewers for their time and diligence in reviewing our submission.  It is clear that each reviewer spent the time to offer helpful suggestions, raise questions, and request clarifications. This will improve our final submission should it be accepted. We address common concerns in a global comment here.

### Evaluating positional encodings for Transformer
GraphTrans can be interpreted as a GNN-based positional encoding for a Transformer. However, alternative positional encodings certainly could outperform the GNN-based embedding. However, due to the permutation equivariance of GNNs, any positional encoding should be carefully chosen.

For ablation, we evaluated the GraphTrans with the sin-cos positional encoding before the Transformer module in the table below, and can observe that it does not help the task performance. If there is an alternative positional encoding reviewers would like to see evaluated, we are excited to benchmark it.

|                           NCI1 | Validation accuracy | Test accuracy |
|-------------------------------:|:-------------------:|:-------------:|
|             GraphTrans (small) |     81.9 ± 1.7%     |  80.2 ± 1.9%  |
| GraphTrans (small) + pos. emb. |     80.5 ± 2.2%     |  79.9 ± 1.9%  |
|             GraphTrans (large) |     84.4 ± 1.6%     |  83.0 ± 1.6%  |
| GraphTrans (large) + pos. emb. |     84.1 ± 1.7%     |  82.3 ± 1.8%  |

|                         NCI109 | Validation accuracy | Test accuracy |
|-------------------------------:|:-------------------:|:-------------:|
|             GraphTrans (small) |     79.8 ± 2.0%     |  79.0 ± 2.5%  |
| GraphTrans (small) + pos. emb. |     80.1 ± 1.9%     |  77.4 ± 2.2%  |
|             GraphTrans (large) |     84.6 ± 2.0%     |  82.5 ± 2.0%  |
| GraphTrans (large) + pos. emb. |     84.2 ± 1.6%     |  82.5 ± 1.5%  |

### Cost of our method (parameter efficiency)
We compare the number of parameters of the GNN baseline and the GraphTrans on different dataset in the table below. Overall, GraphTrans only increases total parameters marginally for Molpcba and NCI. For Code2, GraphTrans is substantially more parameter-efficient than the GNN while improving test F1 score from 0.1629 to 0.1810. One reason for improved parameter efficiency is that the Transformer reduces feature dimensionality before the expensive final prediction layer.

| |  GNN params. | GraphTrans params. | Delta |
| ----------- | :---: | :--------: | :---: |
| Molpcba     | 3.4M  |    4.2M    | 0.8M  |
| NCI         | 0.4M  |    0.5M    | 0.1M  |
| Code2       | 12.5M  |    9.1M     | -3.4M |

---

### Decision · Program_Chairs · 2021-09-27

**Decision:**

Accept (Poster)

**Comment:**

The majority of the reviewers recommend accepting this paper (3 of 4).
The only reviewer not recommending acceptance did not properly engage in discussion and the authors responded to their concerns.
The AC recommends acceptance.